# Molecular Insights into the Thrombotic and Microvascular Injury in Placental Endothelium of Women with Mild or Severe COVID-19

**DOI:** 10.3390/cells10020364

**Published:** 2021-02-10

**Authors:** Arturo Flores-Pliego, Jael Miranda, Sara Vega-Torreblanca, Yolotzin Valdespino-Vázquez, Cecilia Helguera-Repetto, Aurora Espejel-Nuñez, Héctor Borboa-Olivares, Salvador Espino y Sosa, Paloma Mateu-Rogell, Moisés León-Juárez, Victor Ramírez-Santes, Arturo Cardona-Pérez, Isabel Villegas-Mota, Johnatan Torres-Torres, Ángeles Juárez-Reyes, Thelma Rizo-Pica, Rosa O. González, Lorenza González-Mariscal, Guadalupe Estrada-Gutierrez

**Affiliations:** 1Department of Immunobiochemistry, Instituto Nacional de Perinatología, Ciudad de México 11000, Mexico; arturo_fpliego@yahoo.com.mx (A.F.-P.); ceciliahelguera@yahoo.com.mx (C.H.-R.); aurora_espnu@yahoo.com.mx (A.E.-N.); moisesleoninper@gmail.com (M.L.-J.); 2Department of Physiology, Biophysics and Neuroscience, Center for Research and Advanced Studies (Cinvestav), Ciudad de México 07360, Mexico; jamiguz_316@hotmail.com (J.M.); saravegato@gmail.com (S.V.-T.); 3Department of Pathology, Instituto Nacional de Perinatología, Ciudad de México 11000, Mexico; yolotzv@gmail.com; 4Community Interventions Research Branch, Instituto Nacional de Perinatología, Ciudad de México 11000, Mexico; h_borboa1@yahoo.com; 5Clinical Research Branch, Instituto Nacional de Perinatología, Ciudad de México 11000, Mexico; salvadorespino@gmail.com (S.E.yS.); dramateurogell@gmail.com (P.M.-R.); 6Department of Obstetrics, Instituto Nacional de Perinatología, Ciudad de México 11000, Mexico; vhrsan@hotmail.com; 7General Direction, Instituto Nacional de Perinatología, Ciudad de México 11000, Mexico; acardonadr@gmail.com; 8Department of Epidemiology, Instituto Nacional de Perinatología, Ciudad de México 11000, Mexico; isavillegas13@outlook.com; 9Hospital General de México Dr Eduardo Liceaga, Ciudad de México 06720, Mexico; torresmmf@gmail.com (J.T.-T.); dra.angejuarez@gmail.com (Á.J.-R.); thelmarp@yahoo.com (T.R.-P.); 10Department of Mathematics, Autonomous Metropolitan University-Iztapalapa (UAM-I), Ciudad de México 14387, Mexico; lulygo1749@gmail.com; 11Research Division, Instituto Nacional de Perinatología, Ciudad de México 11000, Mexico

**Keywords:** COVID-19, placenta, von Willebrand factor, claudin-5, VE-cadherin, endothelium

## Abstract

Clinical manifestations of coronavirus disease 2019 (COVID-19) in pregnant women are diverse, and little is known of the impact of the disease on placental physiology. Severe acute respiratory syndrome coronavirus (SARS-CoV-2) has been detected in the human placenta, and its binding receptor ACE2 is present in a variety of placental cells, including endothelium. Here, we analyze the impact of COVID-19 in placental endothelium, studying by immunofluorescence the expression of von Willebrand factor (vWf), claudin-5, and vascular endothelial (VE) cadherin in the decidua and chorionic villi of placentas from women with mild and severe COVID-19 in comparison to healthy controls. Our results indicate that: (1) vWf expression increases in the endothelium of decidua and chorionic villi of placentas derived from women with COVID-19, being higher in severe cases; (2) Claudin-5 and VE-cadherin expression decrease in the decidua and chorionic villus of placentas from women with severe COVID-19 but not in those with mild disease. Placental histological analysis reveals thrombosis, infarcts, and vascular wall remodeling, confirming the deleterious effect of COVID-19 on placental vessels. Together, these results suggest that placentas from women with COVID-19 have a condition of leaky endothelium and thrombosis, which is sensitive to disease severity.

## 1. Introduction

Coronavirus infections have ranged from asymptomatic to mild, moderate or severe symptoms in pregnant women. In the Middle East respiratory syndrome coronavirus (MERS-CoV) pandemic, which originated on the Arabian Peninsula in 2012, the pregnancy outcomes remained apparently unaffected [1]. However, during the severe acute respiratory syndrome coronavirus (SARS-CoV) epidemic of 2002–2003 in China, pregnant women had spontaneous miscarriages in the first trimester, delivered preterm, or had pregnancies complicated by intrauterine growth restriction [2]. With the emergence of the novel SARS-CoV-2 and the coronavirus disease 2019 (COVID-19), clinical manifestations of pregnant women with COVID-19 have varied widely from asymptomatic to very severe, and pregnancy complications have included miscarriages [3], fetal distress, premature rupture of membranes, preterm labor, stillbirth [4] and uncontrolled hemorrhage during cesarean delivery [5]. According to the Centers for Disease Control and Prevention in the USA, a higher proportion (12.9%) of preterm live births among women with SARS-CoV-2 infection during pregnancy is present in comparison to the general population (10.2%) [6].

Vertical transmission of SARS-CoV-2 has been suggested, due to positive RT-PCR nasopharyngeal swabs in newborns delivered by cesarean section with neonatal isolation implemented immediately after birth [7,8,9]. Some infants born to mothers with confirmed COVID-19 have displayed problems, including fetal distress, lethargy, vomiting, fever, respiratory distress, thrombocytopenia accompanied by abnormal liver function and even death [9,10,11]. Such symptoms have been displayed in newborns with a SARS-CoV-2 positive and negative RT-PCR test (for review, see [12]), and even with mothers having COVID-19, but with negative RT-PCR tests in amniotic fluid, vaginal secretions, placenta, and breast milk [13].

With the COVID-19 pandemic, interest has arisen concerning the impact of the disease on placental physiology. In pregnant women with COVID-19, placental swabs have been positive for SARS-CoV-2 RNA by RT-PCR in cases where the newborns tested either positive [14] or negative [15,16] for SARS-CoV-2, as in a second-trimester miscarriage with a SARS-CoV-2-negative fetus [3]. The presence of this coronavirus in the placental syncytiotrophoblast (STB) cell layer has been further demonstrated by immunostaining with antibodies against SARS-CoV-2 proteins [17,18] or by RNA in situ hybridization of SARS-CoV-2 spike antigen [19]. Moreover, SARS-CoV-2 virus has been observed invading the human placenta using electron microscopy [20]. However, this observation has been questioned, suggesting that the structures identified are clathrin-coated vesicles and not SARS-CoV-2 virus particles [21].

SARS-CoV-2 cell entry depends on binding of the spike protein organized in trimers to receptor angiotensin-converting enzyme 2 (ACE2) [22]. This receptor is critical, since sequestering ACE2 inside cells due to the loss of Rab7A, a key regulator of endo-lysosomal trafficking, reduces viral entry [23]. SARS-CoV-2 spike protein harbors a multibasic site S1/S2 that undergoes proteolytic cleavage by host proteases like furin, TMPRSS2 and Cathepsin L, which allow the posterior fusion of the viral membrane with a cellular membrane in the endocytic pathway and the release of viral RNA in the cytoplasm of the host cell [22,24].

ACE2 is abundantly present in the lung and small intestine epithelia, as in arterial and venous endothelial cells in all organs [25]. In the human placenta, ACE2 is present in the stromal and perivascular cells of decidua [26], fetal placental vessels [27], and in cytotrophoblast [26]. In the STB cell layer, ACE2 is also present and has been proposed to promote maternal vasodilation through Ang 1–7 release into the maternal circulation [27].

SARS-CoV-2 infects organoids of human blood vessels [28]. Accordingly, viral inclusions have been observed in pulmonary endothelial cells [29], brain [30], transplanted kidney [31], and dermis of patients with COVID-19 chilblain-like lesions [32], where SARS-CoV-2 proteins have also been identified by immunohistochemistry in cutaneous dermal vessels [32,33]. Endothelial cell infection with SARS-CoV-2 is accompanied by a variety of pathological signs, including the accumulation of inflammatory cells, thrombosis, swelling, apoptosis, and pyroptosis [29,31,32,33], a pathway to cell death mediated by caspase-1, which activates the inflammatory cytokines IL-1β and IL-18 (for review, see [34]). These observations suggest that SARS-CoV-2 induces endotheliitis, which could explain the systemic thrombotic and microvascular injury syndrome observed in COVID-19 patients (for review, see [35]). In brain biopsies of patients with COVID-19, no evidence of vasculitis has been found, but thrombotic microangiopathy caused by severe endothelial injury has been observed [36].

Von Willebrand Factor (vWf) plays a critical role in hemostasis as it binds and stabilizes factor VIII in the circulation and mediates platelet–endothelial and platelet–platelet interaction at high shear (for review, see [37]). vWf is assembled as a multimeric protein in endothelial Weibel Palade bodies and is exocyted in response to several stimuli, including inflammatory cytokines. The ultra-large multimers of vWf tether circulating platelets to damaged endothelial sites under high shear stress conditions. In normal circumstances, the metalloproteinase ADAMTS13 cleaves vWf into smaller and less thrombogenic units (for review, see [37]). In thrombotic microangiopathy, like that observed in purpura and other thrombocytopenic conditions like severe sepsis, disseminated intravascular coagulation, and complicated malarial infections, an excess of vWf with a deficiency of ADAMTS13 is observed [38]. Likewise, in patients with severe COVID-19, plasma levels of vWf antigen are increased [39,40,41,42,43,44], while ADAMTS13 activity is normal [42] or diminished [39,40].

Vascular endothelial (VE) cadherin and claudin-5 are adherens and tight junctions (AJ, TJ) proteins, respectively, involved in endothelial cell–cell adhesion and barrier function. VE-cadherin is a classical cadherin present at the adherens junction of endothelial cells, required to prevent the disassembly of blood vessels [45]. Deletion of the VE-cadherin gene leads to early embryonic death associated with severe vascular anomalies [46] and endothelial apoptosis [47]. Likewise, antibodies against VE-cadherin ectodomain block endothelial cell–cell adhesion and increase vascular paracellular permeability [48].

Claudins are major constituents of TJ and are responsible for the ionic selectivity of the paracellular pathway (for review, see [49]). Claudin-5 is specifically present in endothelial cells, and when transfected in L fibroblasts that lack TJs, forms strands that resemble those of endothelial cells where the extracellular, and not the protoplasmic, face of the membrane associates to TJ filaments [50]. In the brain, claudin-5 is required for the establishment of the blood–brain barrier against small molecules (<800 D) [51].

Considering this, we aimed to analyze the impact of COVID-19 in placental endothelium. For this purpose, we have studied the expression of vWf, claudin-5, and vascular endothelial (VE) cadherin in the decidua and chorionic villi of placentas derived from women with mild and severe COVID-19. Here, we found that in the endothelium of decidua and chorionic villi of placentas derived from women with COVID-19, the expression of vWf is increased, being higher in severe cases, suggesting the existence of a thrombotic condition. The altered state of the endothelium in the decidua and chorionic villi of placentas from women with COVID-19 is further confirmed by a decreased expression of both claudin-5 and VE-cadherin in the placentas of women with severe COVID-19, suggesting enhanced vessel permeability. The histological analysis of these placentas revealed thrombosis, infarcts, and remodeling of vascular walls in chorionic villi and decidua, indicating fetal and maternal malperfusion.

## 2. Materials and Methods

### 2.1. Ethics Statement

The study protocol followed the Declaration of Helsinki Ethical Principles for Medical Research Involving Human Subjects. The participants signed the informed consent before their inclusion in this work. The study was approved by the Ethics and Research Internal Review Board of the Instituto Nacional de Perinatología in Mexico City (2020-1-32).

### 2.2. Patient Selection and Specimens

Universal testing with nasopharyngeal swabs and RT-PCR test (La Charité, Berlin protocol) to detect SARS-CoV-2 infection was implemented at the Instituto Nacional de Perinatología for all women who were admitted for delivery, even if asymptomatic; for all positive cases, newborns were tested for SARS-CoV-2 infection using saliva samples.

Placental tissues were obtained immediately after delivery from 11 women with COVID-19 (five mild and six severe) and four control women who delivered by cesarean section, with no evidence of labor. The RT-PCR diagnosis was done during the acute phase of SARS-CoV-2 infection upon admission to delivery. Healthy controls were paired by gestational age. Clinical data and outcomes of enrolled women were obtained from the electronic medical records.

### 2.3. RT-PCR for Placental SARS-CoV-2 Infection

For detection of SARS-CoV-2 RNA in the placenta, the tissue was disrupted through mechanical lysis using a FastPrep instrument (MP Biomedicals, Solon, OH, USA), within 1 h after obtention. Then, RNA was purified using the Direct-zol RNA Miniprep Kit (Cat. R2050; Zymo Research, Irvine, CA, USA). SARS-CoV-2 RNA was detected following La Charité, Berlin protocol [52], using TaqPath 1 step RT-PCR master Mix, CG commercial kit (Cat. A15299 Thermo Fisher Scientific, Waltham, MA, USA), and probes and primers designed for RdRP and E viral genes. RNase P human gene was used as RNA isolation control. RT-qPCR was performed on a StepOnePlus instrument (Applied Biosystems/Thermo Fisher Scientific, Waltham, MA, USA). Each RT-PCR reaction contained an enzyme mix, primers, probes, and RNA (5 µL each one) [52]. Conditions at the thermocycler were set as previously reported [52]. Ct values were collected using threshold at 0.035 fluorescence level.

### 2.4. Immunofluorescence

Placental tissues in paraffin blocks were cut to a thickness of 1 µm, heated overnight at 60 °C and deparaffinized in xylene (Cat. X3s-4; Fisher Scientific, Loughborough, Leicestershire, UK), rehydrated in 100% ethanol (Cat. E-7023; Sigma-Aldrich, St. Louis, MO, USA), 90% ethanol, 70% ethanol, and twice in H_2_O. In the case of VE-cadherin and claudin-5, for epitope retrieval, sections were kept for 40 min in 10 mM citrate buffer at 95 °C. Subsequently, sections used for the detection of vWf were permeabilized with PBS containing 0.5% Triton X-100 for 30 min and incubated in pre-warmed 0.23% (*w*/*v*) pepsin (Cat. P-7000; Sigma-Aldrich) in 0.01 M HCl at 37 °C for 8 min, and then rinsed in distilled H_2_O. These sections were then washed with PBS containing 0.2% Triton X-100 and immunofluorescence buffer (Cat. A3059; Sigma, Poole, Dorset, UK). Alternatively, sections used to detect VE-cadherin and claudin-5 were washed with PBS containing 0.2% Triton X-100 for 10 min. Then, samples were blocked with BSA (immunoglobulin (Ig) free, Cat. 1331-A, Research Organics, Cleveland, OH, USA) for 1 h. Samples were next incubated overnight at 4 °C in a humidified chamber with sheep polyclonal antibodies anti-vWf conjugated with fluorescein isothiocyanate (FITC) (Cat. ab8822; Abcam, Cambridge, MA, USA; dilution 1:100), rabbit polyclonal antibodies anti-claudin-5 (Cat. 34–1600, Invitrogen, Camarillo, CA, USA) and mouse monoclonal antibodies anti-VE-cadherin (Cat. sc-9989, Santa Cruz Biotechnology, Santa Cruz, CA, USA). We also used donkey antibodies coupled to Alexa 647 against mouse IgG (Cat. A31571, Invitrogen), and rabbit IgG (Cat. A31573, Invitrogen). Cell nuclei were evidenced through DNA staining with 300 nM DAPI (4′,6-diamidino-2-phenylindole, dilactate) (Cat. 422801; Biolegend, San Diego, CA, USA) and mounted using Dako Fluorescent mounting medium (Cat. S3023; Dako, Carpinteria, CA, USA). For claudin-5 and VE-cadherin, we also added the autofluorescence quenching kit True View™ (Cat. SP-8400, Vector laboratories, Burlingame, CA, USA), before mounting. Samples were analyzed on an LSM 510 Meta inverted confocal microscope based on an Axiovert 200 M motorized microscope (Carl Zeiss, Oberkochen, Germany) or on an SP8 confocal microscope (Leica, Weitzlar, Germany). The Fiji-ImageJ software (National Institute of Mental Health, Bethesda, MD, USA) [53] was employed to obtain the fluorescence intensity values. For florescence quantification, three random fields per experimental condition were selected, and the figures show representative images of these fields.

### 2.5. Histochemical Staining and Hofbauer Cell Assessment

Placental tissues were fixed with 10% para-formaldehyde and then embedded in paraffin. Sections of 3 μm were cut and stained with hematoxylin and eosin according to standard protocols [54]. Hofbauer cells were identified in these tissues by immunohistochemical analysis, using a rat monoclonal antibody against CD68 (Cat. ab53444, Abcam, San Diego, CA, USA; dilution 1:1000), counterstained with hematoxylin. The number of Hofbauer cells present in the tissue was determined using the analysis of software Zen (version ZEN 2.3 lite, Carl Zeiss Microscopy, Jena, Germany). Fields were selected at 20×, and CD68+ macrophage count was performed at 40× in five different high-power optical fields per placenta.

### 2.6. Statistical Analysis

The three tests of normality of D’Agostino (Skewness, Kurtosis and Omnibus), and the equal variances test of Levene and Bartlett were employed to confirm the assumptions about residuals in One-Way ANOVA. The F test of ANOVA for equal variances and the F test with Welch correction for unequal variances were used, followed by the multiple comparison tests of Bonferroni, Dunnett, or Duncan. The Kruskal–Wallis test and its multiple comparison test were used for not normally distributed data. The legend of each figure indicates the detailed statistical analysis employed. Data are expressed as mean ± SD, and statistical significance was considered for *p* < 0.05.

## 3. Results

Here, we studied the expression of vWf, claudin-5, and VE-cadherin in the decidua and chorionic villi of placentas derived from control and SARS-CoV-2 infected women. The clinical data and outcomes of control women and those with mild and severe COVID-19 are summarized in Table 1, Table 2 and Table 3, respectively.

### 3.1. vWf Is Overexpressed in the Endothelium of Decidua and Chorionic Villi of Placentas Derived from Women with COVID-19 according to Disease Severity

Placentas from control women show a clear immunofluorescence staining for vWf in decidual endothelium (Figure 1a,b) and chorionic villi (Figure 2a,b). In the placenta of women with COVID-19, staining of vWf significantly increases in the decidual and chorionic villi endothelium. The most abundant vWf stain was observed in severe cases. Hence, this result indicates that the placental endothelium of women with COVID-19 displays a characteristic frequently observed in a thrombotic condition.

### 3.2. The Expression of Claudin-5 Diminishes in the Endothelium of Decidua and Chorionic Villi of Placentas from Women with Severe COVID-19

Next, we analyzed the expression of claudin-5 in the endothelium of the decidua and chorionic villi of control women and those with mild or severe COVID-19. By immunofluorescence, we did not observe significant changes in the expression of claudin-5 in decidua (Figure 3a,b) and chorionic villi (Figure 4a,b) of women with mild COVID-19 in comparison to control. However, in placentas of women with the severe form of the disease, we found a significant decrease in claudin-5 expression in decidua (Figure 3a,b) and chorionic villi (Figure 4a,b). Since claudin-5 is the main claudin of endothelial cells [50], these results strongly suggest that TJs in the decidual endothelium and chorionic villi of women with COVID-19 become leaky as the severity of the disease augments.

### 3.3. VE-Cadherin Expression Diminishes in Decidua and Chorionic Villi Endothelium of Placentas from Women with Severe COVID-19

Since VE-cadherin is a crucial protein of the AJs of endothelial cells [45] whose alteration leads to an increase in vascular permeability [48], we next analyzed the expression of this protein in the endothelium of the decidua and chorionic villi. By immunofluorescence, we detected no significant change in VE-cadherin expression in the decidua of women with mild COVID-19 compared to control women. In contrast, VE-cadherin expression in the decidua is significantly reduced in women with severe COVID-19 (Figure 5a,b). A similar pattern is observed in the chorionic microvilli, where the decreased expression of VE-cadherin is detected in women with severe COVID-19 but not in those with mild disease (Figure 6a,b). These results reinforce the observations done with claudin-5, suggesting that COVID-19 augments the paracellular permeability of the endothelium in the chorionic villi.

### 3.4. Placentas of Women with COVID-19 Display Histological Alterations Indicative of Vasculopathy and a Higher Number of Hofbauer Cells Is Observed in Placentas from Women with Severe COVID-19

Since thrombotic and microvascular injury syndrome has been observed in patients with COVID-19 (for review, see [35]), we next analyzed if histological alterations indicative of fetal vascular malperfusion (FVM) or maternal vascular malperfusion (MVM) were present in the placentas of women with COVID-19. The histological analysis revealed the presence in chorionic villi of subacute thrombosis with remodeling of vascular wall, and extensive parenchymal infarcts with intervillositis. In the decidua of women with COVID-19, we found vasculopathy with remodeling of the vascular wall. Instead, placentas from control women had no histological alterations (Figure 7). In addition, to confirm the inflammatory state of placentas of women with COVID-19, we assessed the number of fetal macrophages, known as Hofbauer cells, in the parenchyma of chorionic villi. Figure 8 shows an increase in Hofbauer cells in chorionic villi of women with severe COVID-19, compared to control placentas and tissues from women with the mild form of the disease (** *p* < 0.01).

## 4. Discussion

COVID-19 induces, in some patients, a thrombotic and microvascular injury syndrome triggered by several mechanisms, including a cytokine storm, hypoxic vaso-occlusion, direct activation of immune and vascular cells by virus infection, and the development of pathogenic autoantibodies targeting phospholipids and phospholipid-binding proteins [55] (for review, see [35]).

SARS-CoV-2 has been detected in the placenta of women with COVID-19 [14,15,16,17,18,19], and its receptor ACE2 is present in numerous endothelium [29,30,31,32,33]. Since COVID-19 induces, in some patients, a thrombotic and microvascular injury syndrome (for review, see [35]), here, we have explored if placentas from women with COVID-19 exhibit an altered expression of vWf, claudin-5 and VE-cadherin in the decidua and chorionic villi.

In patients with severe COVID-19, plasma levels of vWf antigen are increased [39,40,41,42,43,44], and in the placenta of healthy women, vWf has been found in endothelium, STB and chorionic villous stroma [56]. In placentas from women with COVID-19 we found that this factor, whose increase may predict an augmented risk of thrombosis [57], is elevated compared to controls in the endothelium of both decidua and chorionic villi, especially in severe cases. This observation suggests that COVID-19 represents a thrombotic risk in human placenta.

Changes in vWf expression have been explored in other pathological conditions of pregnancy. Thus, in pregnancies with intrauterine growth restriction that occurs when a fetus does not reach its growth potential, the expression of vWf is higher, although not at a statistically significant level [58]. In preeclampsia, a higher amount of vWf is found in maternal plasma [56,59], with no differences in the expression in the chorionic villous endothelium and stroma in comparison to normal pregnancies [56]. Others instead reported a decrease in placental vWf in the STB accompanied by an increase in the intervillous space in preeclampsia, thus suggesting injury to the STB cell layer that favors the release of vWf from Weibel–Palade bodies into the maternal space [60].

The cell–cell adhesion complex of endothelial cells constituted by AJ and TJs is critical to prevent vascular leakage and for proper placental perfusion. To test if COVID-19 induced damage to placental vessels, we analyzed the expression of VE-cadherin and claudin-5, the main molecular components of endothelial AJ [45,46,47,48] and TJs, respectively. We found that while there is no significant change in expression of VE-cadherin and claudin-5 in the decidua and chorionic villi of placentas from women with mild COVID-19, the amount of both adhesion molecules decreases significantly in the tissues derived from women with severe COVID-19. Hence, these results indicate that AJ and TJ proteins are sensitive to damage induced by COVID-19, and further reveal that the deleterious effect detected on the apical junctional complex of placental endothelium correlates with COVID-19 severity and the development of inflammation determined by the increased number of Hofbauer cells present in the parenchyma of chorionic villi.

Previous studies have reported the expression of AJ and TJ proteins in chorionic villi of human placenta at term, finding that the STB expresses E-cadherin, ZO-1 and ZO-2, JAM-B, occludin, claudins -1, -3, -4, -5 -7 and -16, while the vessels in the parenchyma of chorionic villi display ZO-1, occludin, JAM-C, and claudins -1, -3, -4 and -5 [61,62,63]. ZO-1, JAM-C, and claudin-5 can be observed in large and small vessels, whereas occludin and claudins 1, -3 and -4 are mainly present in large placental vessels [61,62].

In human placenta, VE-cadherin was previously observed in endothelial cells of the decidua [64] and the chorionic villus [65,66], as well as in the STB, where the expression decreases towards term in normal pregnancies but not in those complicated by preeclampsia [67]. However, in preeclampsia, VE-cadherin expression in chorionic villi endothelial cells was not altered [66]. Diabetes mellitus is another disease linked to endothelial dysfunction (for reviews, see [68,69]), and in insulin-treated gestational diabetes (GDM), a reduction in VE-cadherin expression in chorionic vessels was found [70]. In contrast, placentas of women with diabetes type 1, type 2, and GDM display normal levels of VE-cadherin in the chorionic vessels if not treated with insulin [71].

Previous studies have shown that the amount of claudins -1 and -5 present in Triton X-100 insoluble fractions, which corresponds to claudin associated with the actomyosin cytoskeleton, diminishes in preeclamptic compared to healthy placentas [61], whereas the expression of claudin-4 at the basolateral membrane of the STB diminishes in placentas derived from ZIKV-infected women [62].

Our histological analysis of COVID-19 placentas revealed the presence of thrombosis, infarcts, and vascular wall remodeling in chorionic villi and decidua. These findings are in agreement with previous observations done in placentas from women with COVID-19, showing that the chorionic terminal villi have features of fetal vascular malperfusion including infarctions [14,72], avascular villi [72,73,74], fibrin deposition in fetal vessels [72], intervillous hemorrhages [16] and thrombi [72,73], capillary congestion and focal microchorangiosis [16], a vascular hyperplastic process observed in placental tissue after periods of low-grade hypoxia [75]. Maternal vascular malperfusion has also been reported with decidual arteriophathy, including maternal vessels with artherosis and fibrinoid necrosis as well as hypertrophy of membrane arterioles [72].

We also observed a higher number of Hofbauer cells in placentas from women with severe COVID-19. Similar observations have been done in placental pathologies involving infection and inflammation [76], and, in the case of placentas of women with COVID-19, other studies have reported intervillositis with inflammatory infiltrate of macrophages, T lymphocytes, neutrophils, and monocytes [3,17,18,19].

A preeclampsia-like syndrome has been described in pregnant women with COVID-19, with indistinguishable clinical symptoms and similar histopathological findings [77]. This could be due to the pyroptosis induced by SARS-CoV-2 replication and release [78]. In early preeclampsia, pyroptosis is known to induce the release of alarmins and placental debris into maternal circulation [79], which triggers thrombosis, intramural fibrin deposition, villous stromal-vascular karyorrhexis, and villous infarction [72,73,74]. Altogether, these changes lead to placental dysfunction and fetal growth restriction [80]. In this context, our findings pose a challenge to the differential diagnosis of the hypertensive disease of pregnancy, highlighting the importance of considering the molecular profile of the disease and not only the associated signs and symptoms.

Although placental vasculopathy associated with COVID-19 has been reported previously, it was described in placentas from a heterogeneous population, including women with associated comorbidities (e.g., severe preeclampsia, infection, placenta previa, preterm labor, or diabetes), and vaginal delivery or cesarean section [81,82,83,84]. Instead, our study is more homogeneous, including only women in the acute phase of infection, without associated comorbidities, and having babies delivered by cesarean section. This approach allows us to assume that the placental histological findings and the molecular changes that correlate with these histological observations and with the severity of the disease are due to COVID-19 infection. On the other hand, the limitations of our study are that we were not able to measure circulating vWf in the studied women and that severity markers, such as fibrinogen, D-dimer, and procalcitonin as well as X-rays and CAT, were determined only in women with severe COVID-19.

In conclusion, our study reveals that placentas from women with COVID-19 display a thrombotic and microvascular injury syndrome, including the overexpression of vWf in endothelium coupled with the decreased expression of VE-cadherin and claudin-5 in chorionic villus and decidua that correlate with disease severity. The presence of thrombosis, infarcts, and vascular wall remodeling in chorionic villi and decidua further confirm the deleterious effect of COVID-19 on placental vessels.

## Figures and Tables

**Figure 1 cells-10-00364-f001:**
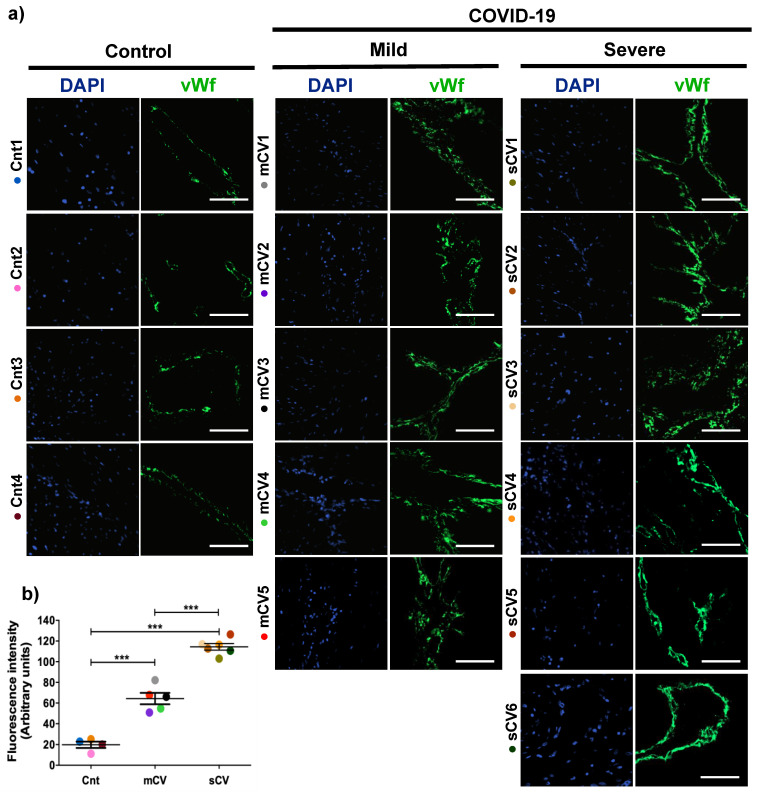
The expression of vWf augments in decidual endothelium of placentas of women with COVID-19. Paraffin blocks of placenta from women with COVID-19 and controls were cut in 1 µm slices, deparaffinized and processed for immunofluorescence with antibodies against vWf. (**a**) Representative images of vWf in decidua. DNA of nuclei was stained with DAPI. Bar, 100 μm. (**b**) Quantification of mean fluorescent intensity done on three independent images from each condition. Data are expressed as mean ± SD; F ANOVA test followed by Duncan’s multiple comparison test, *** *p* < 0.001. Cnt, control; mCV, mild COVID-19; sCV, severe COVID-19.

**Figure 2 cells-10-00364-f002:**
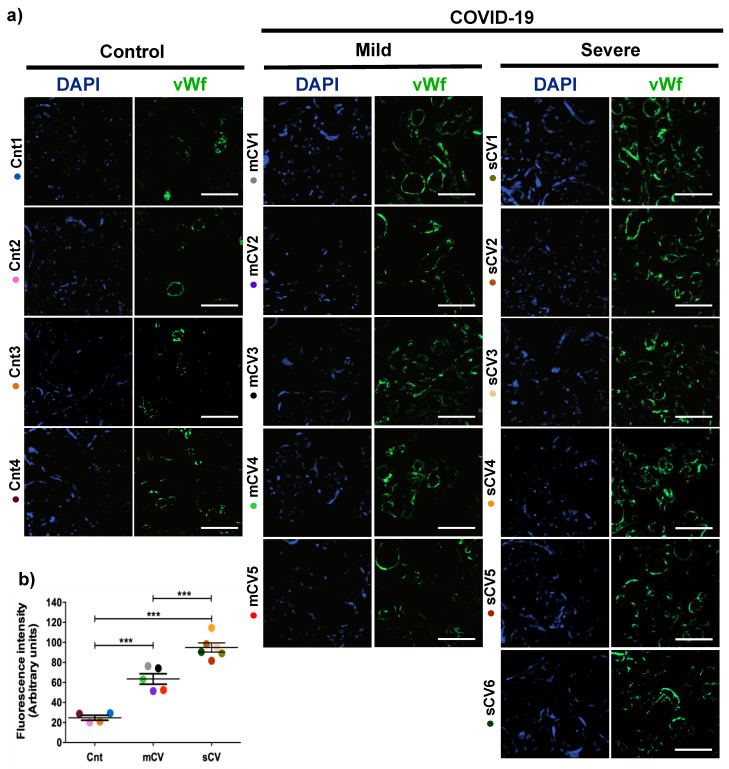
The expression of vWf increases in the endothelium of chorionic villi in placentas of women with COVID-19. Paraffin blocks of placentas derived from women with COVID-19 and controls were cut in 1 µm slices, deparaffinized and processed for immunofluorescence with antibodies against vWf. (**a**) Representative images of vWf in chorionic villi. DNA of nuclei was stained with DAPI. Bar, 100 μm. (**b**) Quantification of mean fluorescent intensity done on three independent images from each condition. Data are expressed as mean ± SD; F ANOVA test followed by Duncan’s multiple comparison test, *** *p* < 0.001. Cnt, control; mCV, mild COVID-19; sCV, severe COVID-19.

**Figure 3 cells-10-00364-f003:**
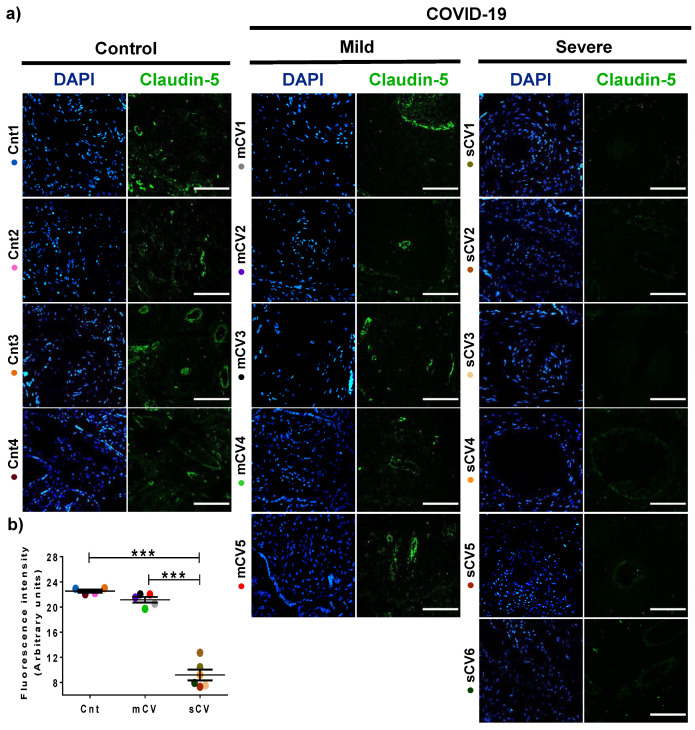
The expression of claudin-5 decreases in decidual endothelium of women with severe COVID-19. Paraffin blocks of placentas derived from women with COVID-19 and controls were cut in 1 µm slices, deparaffinized, and processed for immunofluorescence with antibodies against claudin-5. (**a**) Representative images of claudin-5 in decidua. DNA of nuclei was stained with DAPI. Bar, 100 μm. (**b**) Quantification of mean fluorescent intensity done on three independent images from each condition. Data are expressed as mean ± SD; F test with Welch correction followed by the multiple comparison tests of Bonferroni and Dunnett, *** *p* < 0.001. Cnt, control; mCV, mild COVID-19; sCV, severe COVID-19.

**Figure 4 cells-10-00364-f004:**
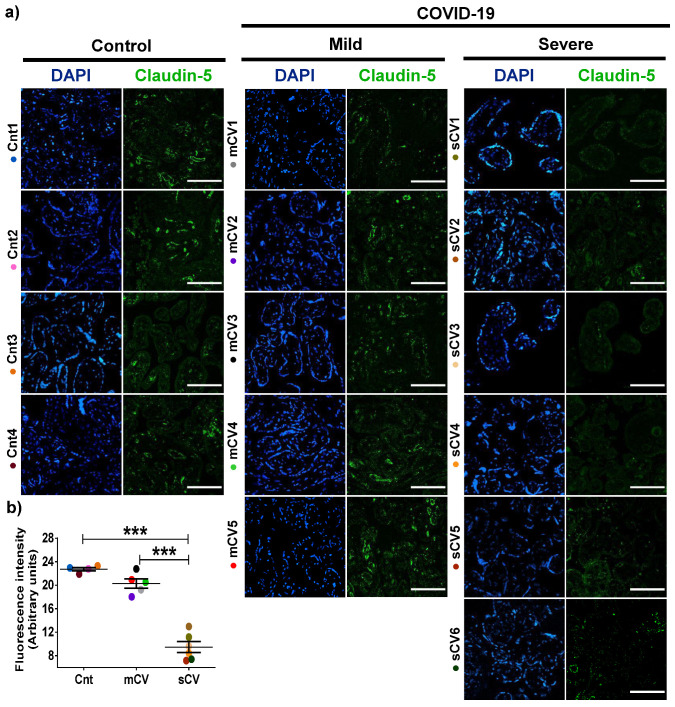
Claudin-5 expression diminishes in the endothelium of chorionic villi of placentas from women with severe COVID-19. Paraffin blocks of placentas derived from women with COVID-19 and controls were cut in 1 µm slices, deparaffinized, and processed for immunofluorescence with antibodies against claudin-5. (**a**) Representative images of claudin-5 in chorionic villi. DNA of nuclei was stained with DAPI. Bar, 100 μm. (**b**) Quantification of mean fluorescent intensity done on three independent images from each condition. Data are expressed as mean ± SD; One way ANOVA F test followed by the multiple comparison tests of Bonferroni and Dunnett, *** *p* < 0.001. Cnt, control; mCV, mild COVID-19; sCV, severe COVID-19.

**Figure 5 cells-10-00364-f005:**
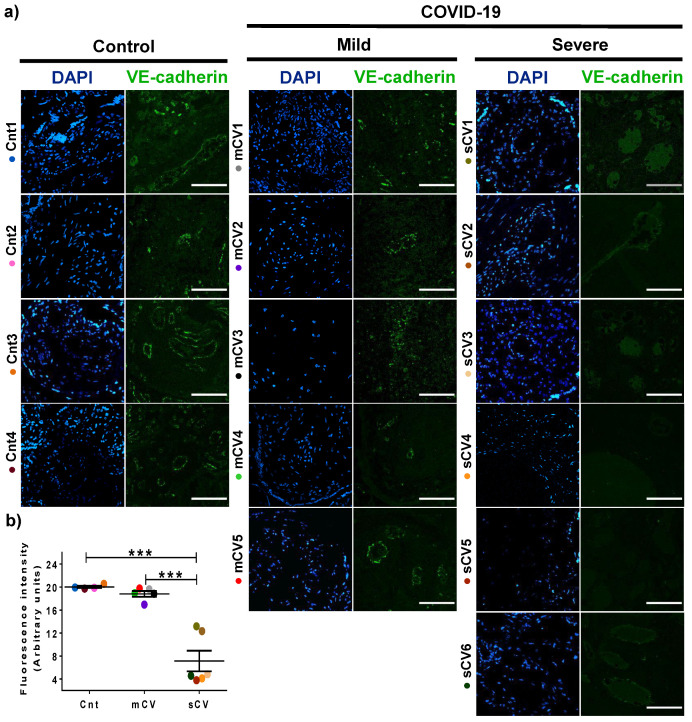
VE-cadherin expression in decidual endothelium decreases in placentas from women with severe COVID-19 but no in those with mild disease. Paraffin blocks of placental tissue derived from women with COVID-19 and controls were cut in 1 µm slices, deparaffinized, and processed for immunofluorescence with antibodies against VE-cadherin. (**a**) Representative images of VE-cadherin in decidua. DNA of nuclei was stained with DAPI. Bar, 100 μm. (**b**) Quantification of mean fluorescent intensity done on three independent images from each condition. Data are expressed as mean ± SD; F test with Welch correction followed by the multiple comparison tests of Bonferroni and Dunnett, *** *p* < 0.001. Cnt, control; mCV, mild COVID-19; sCV, severe COVID-19.

**Figure 6 cells-10-00364-f006:**
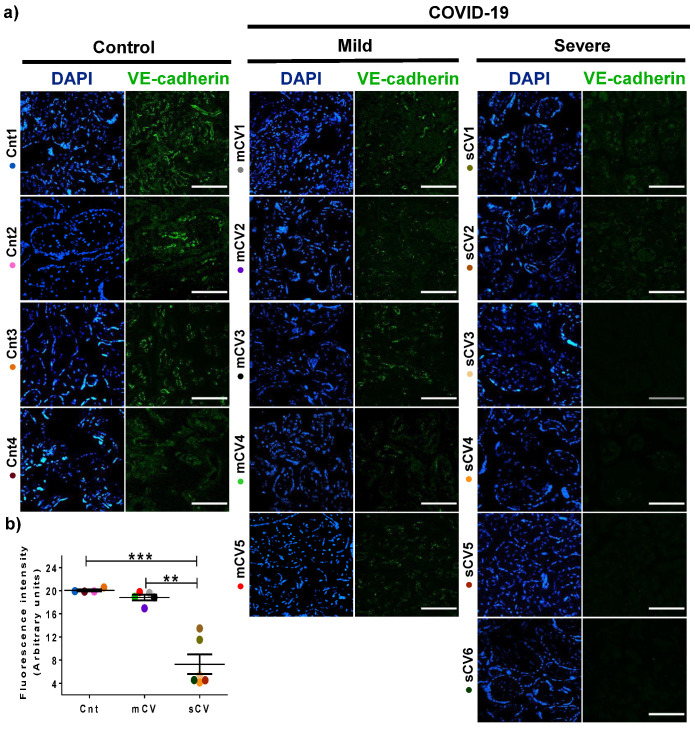
VE-cadherin expression diminishes in the endothelium of chorionic villi of placentas from women with severe COVID-19. Paraffin blocks of placentas derived from women with COVID-19 and controls were cut in 1 µm slices, deparaffinized, and processed for immunofluorescence with antibodies against VE-cadherin. (**a**) Representative images of VE-cadherin in chorionic villi. DNA of nuclei was stained with DAPI. Bar, 100 μm. (**b**) Quantification of mean fluorescent intensity done on three independent images from each condition. Data are expressed as median ± SD; Kruskal–Wallis test and its multiple comparison test, ** *p* = 0.02, *** *p* < 0.001. Cnt, control; mCV, mild COVID-19; sCV, severe COVID-19.

**Figure 7 cells-10-00364-f007:**
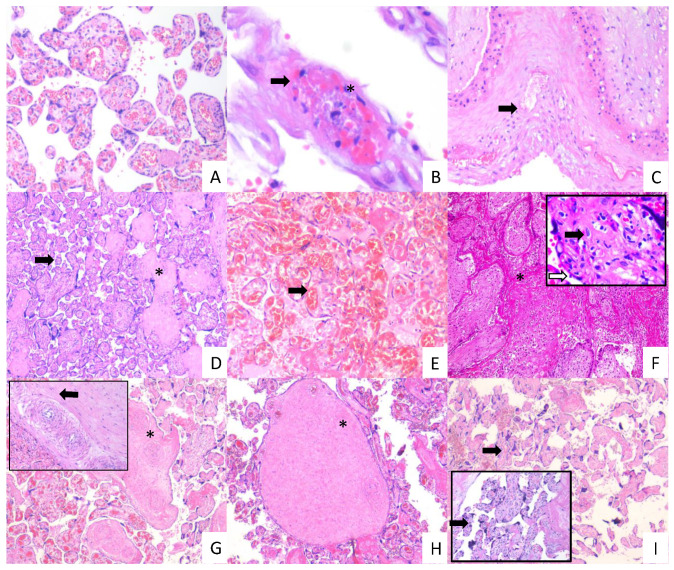
Histopathology of placentas from women with mild and severe COVID-19 shows signs of maternal and fetal vascular malperfusion and decidual vasculopathy. Representative placental sections stained with H&E (3 µm). Villous placental parenchyma from control women with no histological alterations ((**A**); 10×). Placental tissue from women with mild COVID 19 showing decidual vasculopathy characterized by fibrinoid necrosis (arrows), and karyorrhexis (*) ((**B**,**C**); 40× and 10×, respectively), avascular villous (*), and accelerate villous maturation (small villi not expected for gestational age) (arrow) ((**D**); 20×), and chorangiosis (arrow) ((**E**); 20×). Placental tissue from women with severe COVID 19 showing extensive parenchymal infarcts with accumulation of fibrin (*) ((**F**); 10×), acute and chronic inflammation in villous space (villitis, black arrow) and villous (intervillositis, white arrow) (**F**); magnification, 20×); subacute thrombosis of the intermediate villi with remodeling of the vascular wall (*) ((**G**); 10×) and vessel obliteration (arrow) ((**G**); magnification, 20×); avascular villi (*) ((**H**); magnification, 20×), and severe maternal vascular malperfusion with accelerate villous maturation (small villi not expected for gestational age) ((**I**); 10×) and increased syncytial knots (arrows) ((**I**); magnification, 20×).

**Figure 8 cells-10-00364-f008:**
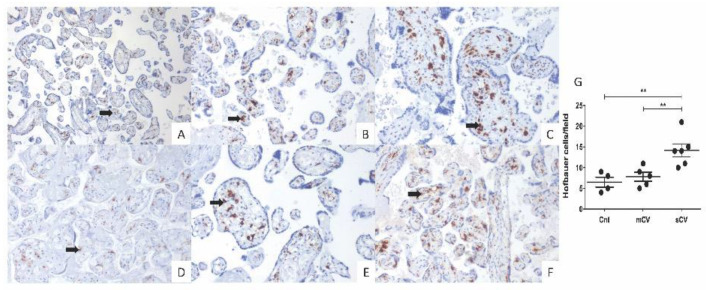
Placental chorionic villi from women with severe COVID-19 display a higher number of Hofbauer cells. Hofbauer cells (arrows) were detected with an antibody against CD68 and counterstained with hematoxylin. Representative 3 µm placental sections (20×) from control women (**A**,**D**), women with mild COVID-19 (**B**,**E**), and women with severe COVID-19 (**C**,**F**). (**G**) Quantification of Hofbauer cells in placentas from control women (Cnt), with mild COVID-19 (mCV), and severe COVID-19 (sCV). The number of Hofbauer cells was obtained by counting CD68+ positive cells in five optical fields per placenta. Each dot represents the mean of Hofbauer cells in five fields per placenta of each condition. Data are expressed as mean ± SD; F ANOVA test followed by Duncan’s multiple comparison test, ** *p* < 0.01.

**Table 1 cells-10-00364-t001:** Clinical data of control women and their newborns.

	Cnt1	Cnt2	Cnt3	Cnt4
Maternal Age (years)	24	17	17	20
Maternal comorbidity	None	None	None	None
Maternal weight (kg)	NA	53.7	62.1	65.0
Maternal height (m)	NA	1.67	1.64	1.59
GA at Diagnosis (weeks) GA at Delivery	13.6	38.1	40.2	30.3
(weeks)	13.6	38.1	40.2	30.3
Delivery mode	Curettage	CS	CS	CS
NB weight (g)	NA	2315	3170	1320
NB Gender	NA	Male	Female	Female
NB weight classification	NA	SGA	AGA	AGA
White blood cell count/UL	7500	6600	10,500	11,300
Lymphocyte count (%)	9.0	15.1	10.1	9.8
Platelets cells/UL	187,000	236,000	274,000	167,000
Thrombin time (s)	11.0	9.5	10.9	10.8
Prothrombin time	26.9	24.9	36.2	29.6

NA, Not Available; NB, Newborn; CS, Cesarean Section; AGA, Appropriate for Gestational Age; SGA, Small for Gestational Age.

**Table 2 cells-10-00364-t002:** Clinical data of women with mild COVID-19 and their newborns.

	mCV1	mCV2	mCV3	mCV4	mCV5
Maternal Age (years)	28	20	34	19	20
Maternal comorbidity	None	None	Epilepsy	None	None
Maternal weight (kg)	NA	65.4	93.7	75.0	57.0
Maternal height (m)	NA	1.69	1.67	1.59	1.50
COVID-19 stage	Acute	Acute	Acute	Acute	Acute
PCR Mother	+	+	+	+	+
PCR NB/Fetal	+	+	+	+	+
PCR Placenta	−	−	−	−	−
COVID-19 symptoms	Cough Headache	None	None	None	None
GA at Diagnosis (weeks) GA at Delivery	13.0	40.3	39.5	33.4	26.4
(weeks)	13.0	40.4	39.5	33.5	26.6
Delivery mode	Curettage	CS	CS	CS	CS
NB weight (g)	NA	3410	3615	1946	978
NB Gender	NA	Male	Male	Male	Female
NB weight classification	NA	AGA	LGA	AGA	AGA
White blood cell count/UL	NA	8400	7900	10,200	107,700
Lymphocyte count (%)	NA	30.1	17	2.9	11
Platelets cells/UL	NA	224,000	208,000	177,000	341,000
Thrombin time (s)	NA	9.2	10.3	10.8	10.3
Prothrombin time	NA	29.8	23.8	28.6	30.4

mCV, mild COVID-19; NA, Not Available; NB, Newborn; CS, Cesarean Section; AGA, Appropriate for Gestational Age; LGA, Large for Gestational Age.

**Table 3 cells-10-00364-t003:** Clinical data of women with severe COVID-19 and their newborns.

	sCV1	sCV2	sCV3	sCV4	sCV5	sCV6
Maternal Age (years)	37	25	25	37	36	39
Maternal comorbidity	None	None	None	None	None	None
Maternal weight (kg)	91	67	66	60	75.5	77.5
Maternal height (m)	1.65	1.55	1.60	1.50	1.60	1.64
COVID-19 stage	Acute	Acute	Acute	Acute	Acute	Acute
PCR Mother	+	+	+	+	+	+
PCR NB/Fetal	−	−	−	−	−	−
PCR Placenta	−	−	−	−	−	−
COVID-19 symptoms	Dyspnea, myalgias, arthralgia, diarrhea	Cough, fever, dyspnea, myalgias, arthralgias, rhinorrhea	Cough, fever, dyspnea, myalgias, arthralgias, diarrhea, rhinorrhea	Cough, fever, myalgias, arthralgias	Cough, fever, dyspnea	Cough, fever, dyspnea
GA at Diagnosis (weeks)	27.6	34.6	28.0	38	39.1	39.1
GA at Delivery (weeks)	27.6	34.6	28.0	38	39.1	39.1
Delivery mode	CS	CS	CS	CS	CS	CS
NB weight (g)	1600	2200	1250	2640	2330	2900
NB Gender	Female	Female	Female	Female	Female	Male
NB weight classification	AGA	AGA	AGA	AGA	SGA	AGA
White blood cell count/UL	11,200	8900	16,500	7600	9900	8800
Lymphocyte count (%)	16.3	9.5	4.7	24.2	24.1	24.0
Platelets cells/UL	365,000	218,000	347,000	327,000	275,000	232,000
Thrombin time (s)	10.8	9.6	10.0	10.9	13.6	16.4
Prothrombin time	30.8	30.7	22.6	21.5	24.7	23.0
Aspartate aminotransferase (U/L)	28	48	28	22	10	28
Alanine aminotransferase (U/L)	8	24	8	14	10	56
Creatinine (mg/dl)	0.52	1.14	0.53	0.6	0.49	0.64
Fibrinogen (mg/dL)	681	485	856	479	498	601
D-dimer (ng/mL)	1267	1346	1739	3500	5993	4716
Procalcitonin (ng/mL)	0.12	1.72	0.28	1.18	0.02	0.05
Rx or CAT (COVID signs)	+	+	+	+	+	+
Orotracheal intubation	+	−	+	−	−	−
Supplemental O_2_	−	+	−	+	+	+

sCV, severe COVID-19; NB, Newborn; CS, Cesarean Section; AGA, Appropriate for Gestational Age; Rx, X-ray; CAT, Computerized axial tomography.

## Data Availability

The data presented in this study are available on request from the corresponding author. The data are not publicly available due to privacy.

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
