# Peer review of "Molecular Insights into the Thrombotic and Microvascular Injury in Placental Endothelium of Women with Mild or Severe COVID-19"

_cells, 2021, doi:10.3390/cells10020364_

Round 1

Reviewer 1 Report

In this manuscript, the authors had tried to analyze the impact of COVID-19 in placental endothelium. They had looked into the expression of vWf, Claudin-5, and VE in the decidua and chronic villi of placenta-derived from COVID 19 patients. Their hypothesis was that the enhanced vessel permeability resulting in thrombosis, infarcts, and remodeling of vascular walls in chorionic villi and decidua, indicative of fetal and maternal malperfusion were associated with patients with COVID-19.

Comments

  1. The authors had recruited 4 controls and 8 COVID 19 positive patients. The sample size was very small to make any prediction or observation. Appropriate power analysis was required for proper sample size.
  2. No statistical method had been described in the manuscript.
  3. IHC images in Figure 1 were not clear. Higher magnification was expected.
  4. The Claudin 5 staining had high background and the difference between mild and severe staining was not convincing.
  5. As the authors wanted to predict that these changes are due to COVID infection, colocalization with some inflammation, T-Cell and/or Macrophage markers would have established their claim.
  6. Localization and quantification of ACE2 in the tissues were essential for the study.
  7. Placental vascular abnormality associated with COVID-19 had been previously reported. Hence, the novelty of this study was not clear to me.

Reviewer 2 Report

In the original paper entitled „Molecular insights into the thrombotic and microvascular injury in placental endothelium of women with mild or severe COVID-19” by Flores-Pliego and Miranda et al., the Authors analyzed the effects of COVID-19 on placenta endothelium measured by a set of immunofluorescence techniques. Undoubtedly, this is a hot topic, potentially interesting for the readers, however I have a couple of major and minor comments which are listed below.

Major comments

  1. A serious limitation of the study is a low sample size. The Authors studied 5 mild and 3 severe cases of women with COVID-19 and 4 controls. What was a rationale to analyse these numbers in all the groups? Did the Authors estimate the sample size or did they just analysed all available patients consecutively? What inclusion and exclusion criteria to the groups were selected, if any? Actually, in this form, the work can be regarded as a series of case reports.
  2. A description of statistical analysis is missing in Materials and Methods. The Authors give this information in the Figure legends only but they use wrong tests. Student t-test is designed to analyse differences between two groups. In a case of more groups (3 in this case), Bonferroni correction or ANOVA should be used. Furthermore, when taking into account that a number of observation is very low, there is a high probability that the results are not normally distributed. If so, non-parametric tests should be performed. I believe that having extremely low number of patients, the Authors should pay special attention to select the reliable statistical tools. Otherwise, the risk of bias is very high.

Minor comments

  1. Citing 83 references (51 in the Introduction) in a original paper is untypical. I would suggest to keep this number up to 60.

Round 2

Reviewer 1 Report

The authors have addressed every comment and I do not have any other concerns.

Reviewer 2 Report

The Authors responded to my comments. The statistical analysis has been improved. I have no further comments.